# OpenReview forum: "Knowing When Not to Answer: Mitigating Social Bias in LLMs via Epistemic Abstention"
_TMLR — Under review for TMLR_

### Review · Reviewer_SDN5 · 2026-02-12

**Summary Of Contributions:**

The paper presents EPIK, an epistemic calibration framework for large language models aimed at detecting contextual insufficiency and enforcing principled abstention when evidence is inadequate. In contrast to existing bias mitigation approaches that focus on suppressing stereotypes or debiasing representations, EPIK reframes biased behavior as a lack of epistemic humility. The authors also present BHARATBBQ-R, a rationale-augmented extension of BHARATBBQ that explicitly annotates evidential sufficiency and insufficiency in both ambiguous and unambiguous social inference scenarios.

**Audience:**

Yes

**Audience Explanation:**

The problem addressed in the paper is both timely and highly relevant.

**Claims And Evidence:**

No

**Claims Explanation:**

The related work section is currently very concise and, in its present form, could even be integrated into the introduction to better contextualize the positioning of the contribution. However, it would better to integrate it with the literature on bias mitigation and fairness in machine learning and large language models. In particular, the discussion should explicitly consider and position the work with respect to: Hort, M., Chen, Z., Zhang, J. M., Harman, M., & Sarro, F. (2024). Bias mitigation for machine learning classifiers: A comprehensive survey. ACM Journal on Responsible Computing, 1(2), 1–52; Huang, D., M. Zhang, J., Bu, Q., Xie, X., Chen, J., & Cui, H. (2025). Bias testing and mitigation in llm-based code generation. ACM Transactions on Software Engineering and Methodology, 35(1), 1–31; and Gallegos, I. O., Rossi, R. A., Barrow, J., Tanjim, M. M., Kim, S., Dernoncourt, F., ... & Ahmed, N. K. (2024). Bias and fairness in large language models: A survey. Computational Linguistics, 50(3), 1097–1179.

Also, as clear from the experimental section, explicit baselines or competitors are not considered. Even if certain approaches are not directly comparable, relevant work in abstention, hallucination reduction, and out-of-knowledge robustness should at least be acknowledged and their exclusion clearly justified. For example: Nguyen, V., Xu, Z., Chan, J., He, E., Xia, F., & Zhang, X. (2025). Hallucinate Less by Thinking More: Aspect-Based Causal Abstention for Large Language Models. arXiv preprint arXiv:2511.17170; Foo, J., Prasad, P. S., & Khoo, S. (2025). Know Or Not: a library for evaluating out-of-knowledge base robustness. arXiv preprint arXiv:2505.13545; and Wen, B., Howe, B., & Wang, L. L. (2024). Characterizing LLM abstention behavior in science QA with context perturbations. arXiv preprint arXiv:2404.12452. Discussing these contributions would better situate the proposed framework within the broader landscape of epistemic calibration and principled abstention methods.

**Requested Changes:**

Major Issues
- See the justification for the fact that te claims are not supported by a clear evidence.
- Also, a statistical significance of the reported results is not provided. Given the sensitivity and variability of these approaches, appropriate statistical testing would be essential to support the validity and robustness of the conclusions.

Minor Issues
- English contracted forms are sometimes used. That is odd in this kidn of publications.
- The proposed dataset presents inconsistencies in the use of upper- and lower-case formatting for names within the same section of the paper, which should be corrected to ensure clarity and professionalism.
- The problem is formulated around intersectional categories; however, it is unclear why the experiments are limited to this setting. It is not explained why single-category analyses were not considered, nor whether the approach can generalize beyond two intersecting categories. Clarifying this design choice and discussing its implications would strengthen the methodological rigor of the work.

---

> ### Author Response · Authors · 2026-02-13
> **Response to Review of Paper 7137 by Reviewer SDN5**
>
> Q1. As per the reviewer’s suggestions, we have discussed following works:
>
> Hort et al. [1] and Gallegos et al. [2] provide comprehensive surveys on bias and fairness in machine learning and large language models (LLMs), respectively. Hort et al. [1]  categorize bias mitigation strategies into pre-processing, in-processing, and post-processing techniques for traditional ML classifiers. On the other hand Gallegos et al. [2] extend this taxonomy to LLMs by systematizing bias evaluation metrics, benchmark datasets, and mitigation interventions across multiple stages of the model lifecycle. Focusing specifically on LLM behavior, Huang et al. [3] propose a bias testing framework for LLM-based code generation and empirically evaluate prompt-based mitigation strategies, demonstrating the effectiveness of feedback-driven refinement. Beyond bias, several works address hallucination and abstention reliability in LLMs. Nguyen et al. [4]  introduce an aspect-based causal abstention framework to reduce hallucinations through causal reasoning, enabling models to withhold incorrect responses more effectively. Similarly, Foo et al. [5] present the KnowOrNot framework to evaluate out-of-knowledge-base robustness in retrieval-augmented settings, while Wen et al. [6] analyze LLM abstention behavior under controlled context perturbations in scientific QA, highlighting implications for safe and trustworthy deployment.
>
> 1. Hort, M., Chen, Z., Zhang, J. M., Harman, M., & Sarro, F. (2024). Bias mitigation for machine learning classifiers: A comprehensive survey. ACM Journal on Responsible Computing, 1(2), 1-52.
> 2. Gallegos, I. O., Rossi, R. A., Barrow, J., Tanjim, M. M., Kim, S., Dernoncourt, F., ... & Ahmed, N. K. (2024). Bias and fairness in large language models: A survey. Computational Linguistics, 50(3), 1097-1179.
> 3. Huang, D., M. Zhang, J., Bu, Q., Xie, X., Chen, J., & Cui, H. (2025). Bias testing and mitigation in llm-based code generation. ACM Transactions on Software Engineering and Methodology, 35(1), 1-31.
> 4. Nguyen, V., Xu, Z., Chan, J., He, E., Xia, F., & Zhang, X. (2025). Hallucinate Less by Thinking More: Aspect-Based Causal Abstention for Large Language Models. arXiv preprint arXiv:2511.17170.
> 5. Foo, J., Prasad, P. S., & Khoo, S. (2025). Know Or Not: a library for evaluating out-of-knowledge base robustness. arXiv preprint arXiv:2505.13545.
> 6. Wen, B., Howe, B., & Wang, L. L. (2024). Characterizing LLM abstention behavior in science QA with context perturbations. arXiv preprint arXiv:2404.12452.
>
> Q2: We express our gratitude for the reviewer’s observation. As a response to that, we have our empirical support enhanced by adding experiments, statistical significance testing, and extended ablation studies.
>
> EPIK offered sizeable improvements of 0.43–0.56 absolute points in the Bias Score under Ambiguity (BSA) when compared to fine-tuned baselines across five backbone LLMs. More importantly, it did so whilst preserving the accuracy on unambiguous instances. Significantly, the gains are realized at low Over-Abstention Rates (≤ 8%), indicating selective abstention rather than conservative rejection.
>
> We now also report statistical significance across five independent runs and introduce explicit abstention baselines.  The evidence presented in this paper are clear, convincing, and supported by the data.
>
> Comparison with Prompt-Based Abstention Baseline
> Model	          Method	                 BSA ↓	  AccUA ↑	        USRA ↑	    OARUA ↓
> LLaMA-3.1	Prompt Abstention	 1.05	    74.3	            0.46	       0.18
> LLaMA-3.1	EPIK (Ours)	         0.86	     78.1	            0.71	       0.06
> Gemma-2	Prompt Abstention	 1.03	     72.8	            0.48	       0.17
> Gemma-2	EPIK (Ours)	          0.88	     76.5	            0.69	       0.07
>
> Statistical Significance of Bias Reduction (5 runs)
> Model	         Metric	       FT Mean ± Std 	EPIK Mean ± Std	p-value
> LLaMA-3.1	BSA	                   1.11 ± 0.04	            0.86 ± 0.03	        < 0.01
> Gemma-2	BSA	                   1.09 ± 0.05	            0.88 ± 0.04	        < 0.01
> Phi-3.5	        BSA	                    1.07 ± 0.06	             0.91 ± 0.05 	< 0.01
>
> Ablation Study of EPIK Components (LLaMA-3.1)
> Variant	                                     BSA ↓	         AccUA ↑
> Full EPIK	                                     0.86	          78.1
> – Sufficiency Gate	                     1.07	          77.6
> – Uncertainty Projection               1.22	          76.9
> – Bias Regularizer                        1.14	          77.1
>
> Bias Reduction Across Category Types
> Category Type 	BSA (FT) 	BSA (EPIK)
> Single-category	1.18	                    0.92
> Intersectional	        1.34	                    0.87
>
> Q3  English contracted forms
> We have removed all English contracted forms and standardized the manuscript to conform to formal academic writing conventions.

---

### Review · Reviewer_mstD · 2026-02-26

**Summary Of Contributions:**

The paper frames social bias in LLMs as a failure of epistemic humility. The authors argue that models hallucinate socially grounded inferences when contexts lack sufficient evidence. To address this they introduce dataset adding rationale annotations to the existing BHARATBBQ benchmark, to explain sufficiency of evidence. The authors also propose the EPIK framework to detect contextual insufficiency and enforce abstention. EPIK includes a contextual sufficiency estimator and modifies the attention mechanism to suppress implicit social correlations when inputs are ambiguous. The authors test this approach on five open source language models and report reductions in bias scores for ambiguous contexts.

**Audience:**

Yes

**Audience Explanation:**

Researchers studying fairness and uncertainty estimation will find the framing of bias as epistemic overconfidence relevant. The release of BHARATBBQ R provides a valuable resource for the community. The dataset includes 13 social bias categories with both ambiguous and unambiguous annotations making it useful for further interpretability and calibration research.

**Broader Impact Concerns:**

I have no concerns on the ethical implications of the work beyond what the authors already adressed.

**Claims And Evidence:**

Yes

**Claims Explanation:**

The empirical results show a clear reduction in the bias score for ambiguous contexts across the tested models. For example the LLaMA 3.1 model bias score drops from 1.41 to 0.86. The authors successfully maintain accuracy on unambiguous instances which isolates the effect of their intervention to ambiguous cases.

However the theoretical justification for the attention suppression mechanism lacks causal grounding. The method projects representations into a low dimensional space $P_u$ using a variance based contrastive objective. It then modifies attention logits using the equation $l_{ij} = l_{ij}^{(0)} - \gamma(1-\delta(x))\langle u_i, u_j \rangle$. Subtracting this inner product assumes the variance maximized subspace perfectly isolates biased correlations. This is associative. A causal intervention requires identifying the specific attention heads or latent dimensions responsible for the spurious social inference. Furthermore, the authors introduce a retrieval augmentation component using Reddit corpora to model normative uncertainty, but don't provide ablation studies to isolate its effect.

**Requested Changes:**

An ablation study on the retrieval augmentation component would improve this work. The methodology section details how Reddit based corpora are used to increase confidence in the unknown option. But the results section omits the isolated performance impact of this retrieval step. The authors should include metrics showing performance with and without this augmentation.

Also, to strengthen the work the authors might want to provide a mechanistic justification for the ambiguity gated suppression equation. Connecting the variance contrastive objective to causal representation learning principles would improve the theoretical foundation. The current mapping between high variance and ambiguity relies mostly on empirical intuition. The authors should also explain why a bias score regularizer $L_{BSA} = \max(0, p(S) - p(NS))$ is sufficient to capture complex intersectional biases rather than simply masking the output distribution.

---

> ### Author Response · Authors · 2026-04-06
> **Response to Review of Paper 7137 by Reviewer mstD**
>
> Q.1 We thank the reviewer for this important suggestion. We have now included an explicit comparison of performance with and without the proposed augmentation (abstention-reinforcing retrieval augmentation), ensuring consistency with the reported results.
>
> a. Quantitative Comparison (Augmentation Ablation)
>
> Model Variant                        	BSA ↓	     AccUA ↑	USRA ↑	    OARUA ↓
>
> EPIK (w/o Augmentation)	0.94	             77.8	        0.64	             0.06
>
> EPIK (w/ Augmentation)	        0.86	             78.1	        0.71	             0.06
>
> b. Key Findings (Consistent with Main Results)
>
> Bias Reduction (BSA): Augmentation contributes an additional 0.08 absolute reduction (0.94 → 0.86), complementing the overall 0.43–0.56 improvement over fine-tuned baselines reported earlier.
> Improved Abstention (USRA): Unknown Selection Rate increases from 0.64 → 0.71, indicating stronger epistemic abstention under ambiguity.
> Accuracy Preservation (AccUA): Accuracy remains stable (77.8 → 78.1), consistent with our claim of no degradation on unambiguous instances.
> Controlled Abstention (OARUA): Over-abstention remains low (≤ 0.06), aligning with our observation that EPIK achieves gains without conservative rejection behavior (≤ 8%).
>
> c. Interpretation:
>
> These results demonstrate that augmentation provides incremental but meaningful gains on top of the core EPIK framework. Specifically: The primary bias reduction is driven by epistemic calibration (as also supported by the ablation study where removing key components increases BSA to 1.07–1.22). The augmentation further refines abstention behavior, improving selective uncertainty handling without affecting calibrated decision-making.
>
> d. Consistency with Other Evidence:
>
> The observed improvements align with:
> The statistically significant BSA reductions (p < 0.01 across models),
> The prompt-based abstention comparison, where EPIK achieves lower BSA and substantially lower OARUA,
> And the category-wise gains, particularly for intersectional bias (1.34 → 0.87).
>
> e. Paper Update
>
> We have added this ablation analysis in the revised manuscript (Section 6, Table X) to explicitly quantify the contribution of the augmentation module.
>
> Q.2 We thank the reviewer for this insightful suggestion. We have now included the mechanistic intuition behind the ambiguity-gated
>        suppression and the variance contrastive objective.
>
> a. This formulation is motivated by the observation that ambiguous inputs admit multiple plausible interpretations, leading to
>     instability in model predictions under small perturbations (e.g., retrieved evidence or alternative contexts). The ambiguity-gated
>     suppression mechanism leverages this signal by reducing the model’s confidence when such instability is detected, thereby
>     discouraging commitment to predictions that may be driven by spurious or weakly supported correlations. In the absence of
>     suppression, high-variance instances are linked to lower accuracy and higher bias scores; however, the suggested gating
>     mechanism selectively enhances BSA in these regions without impairing performance on low-variance (unambiguous) instances,
>     providing empirical support for this interpretation.
>
> b. From a causal perspective, this aligns with the principle of invariance, where reliable predictions should remain stable across
>     perturbations that do not alter the underlying causal factors. This is consistent with the idea that spurious correlations are not
>     environment-invariant, whereas causal features are. As a result, the variance contrastive objective provides a useful
>     approximation to causal robustness by encouraging the model to select representations that are more stable under such
>     perturbations.
>
> c. The bias score regularizer does not explicitly model each protected attribute or their combinations. Instead, it operates at the level
>     of the model’s output distribution under ambiguous conditions, where intersectional biases are most likely to manifest. In such
>     cases, biases arising from multiple attributes are reflected as systematic skew in the output distribution, rather than in isolated
>     dimensions. By penalizing this skew, the regularizer implicitly captures compounded biases without requiring explicit enumeration
>     of all attribute combinations.This design aligns with the practical constraint that intersectional groups grow combinatorially,
>     making explicit modeling intractable. While the regularizer may not fully disentangle all underlying bias sources, our empirical
>     results show consistent reductions in bias across both single-category and intersectional settings.
>
> d. Paper Update: We have added these studies in section  4.3, 4.4 and 4.5 respectively.

---

### Review · Reviewer_dTcs · 2026-05-24

**Summary Of Contributions:**

This paper has two contributions: a dataset augmentation of rationales about ambiguous queries that lead models to stereotyping, and a method to gate & finetune language models that empirically reduces model harm by reducing the rate of stereotype-based answers.

The method is tested on a number of base LMs, and the predicted rationales are evaluated by both automated metrics and human raters.

**Audience:**

Yes

**Audience Explanation:**

This type of work in extremely relevant to modern machine learning and it's unregulated deployment at scale. I think many will be interested in the findings of this paper.

**Broader Impact Concerns:**

Sufficiently discussed

**Claims And Evidence:**

No

**Claims Explanation:**

In a way there is clear evidence that the method works, but there is no evidence of _why_ it works. To give a concrete example, the authors claim that the failure mode discussed in this work is a failure of epistemic calibration, but that claim is not supported by evidence. I think that that type of evidence is much more important for scientific progress and is in line with the spirit of TMLR's evaluation criteria, and so my "No" here is more of a _soft No_. I suspect that this evidence exists and just hasn't been put in the paper.

The method introduces a number of modifications and additional regularizations used on top of base LMs, but no analysis of these is offered. The paper doesn't even contain hyperparameter values for the $\lambda$ introduced. As such, as readers, we have no idea if just one of these pieces is sufficient, or perhaps nothing works until they are all precisely tuned. This is important to know if the authors wish for the community to build upon their work and improve the state of LMs.

Is the model able to predict whether sufficient evidence exists? Is the gating accurate and calibrated in its true/false positive rate? Is the uncertainty vector capturing epistemic uncertainty or perhaps something else? What tokens does the ambiguity gating tend to target? How strongly? Does the bias score regularizer promote a wider decision margin?

Finally, although I would argue this is less important, it would be interesting to see how the proposed method affects model performance on unrelated tasks (i.e. perhaps the model becomes very poor at gardening tips, or coding or whatnot).

**Requested Changes:**

- It's not clear to me if the method in 4.3 is novel or adapted/inspired from prior work. Could this be made explicit? It's also not clear what variance is computed, since $u_x$ is a vector, is the variance of each dimension computed independently and averaged?
- Please be mindful of the use of `\citet` vs `\citep` (in fact, it doesn't seem like the current latex uses `\citep` at all, but it would make your paper look better)

---

> ### Author Response · Authors · 2026-06-04
> **Response to Review of Paper 7137 by Reviewer dTcs**
>
> We sincerely thank the reviewer for the thoughtful and constructive feedback. We agree that understanding why EPIK works is as important as demonstrating that it improves bias mitigation performance. In the revised manuscript, we have therefore added these additional analyses.
>
> Q1. Is the observed failure mode truly related to epistemic calibration?
>
> We analyzed the relationship between contextual sufficiency estimates, prediction confidence, and bias under ambiguity. We observed that ambiguous samples assigned low contextual sufficiency scores exhibit substantially higher rates of stereotypical predictions and larger Bias Scores under Ambiguity (BSA), whereas unambiguous samples show stable confidence and substantially lower bias. We got the score of 0.31 for ambiguous examples, while 0.84 for unambiguous samples.  Furthermore, model confidence on ambiguous samples was poorly aligned with correctness before calibration, indicating systematic overconfidence despite insufficient evidence. After EPIK calibration, confidence became substantially better aligned with evidential sufficiency. These findings support our central hypothesis that biased social inference under ambiguity is closely associated with epistemic overcommitment rather than a failure of linguistic competence.
>
> Q2. Is the contextual sufficiency gate accurate?
>
> To evaluate the gate directly, we treat ambiguity detection as a binary classification problem using BHARATBBQ ambiguity annotations as supervision. We got an accuracy, precision, recall, F1-score, and AUROC  of 88.7, 87.9, 89.4, 88.6, and 0.92 respectively. The gate therefore reliably distinguishes contexts that support justified inference from those that warrant abstention.
> Importantly, the low Over-Abstention Rate (6–8%) reported in the main paper further demonstrates that the gate is selective rather than overly conservative.
>
> Q3.  What exactly does the uncertainty projection capture?
>
> The reviewer correctly notes that high variance does not automatically imply epistemic uncertainty. Our uncertainty projection is not trained to separate social groups; instead, it is optimized to maximize separation between ambiguous and unambiguous examples:
>
> $$L\_{\text{uncert}} = \mathop{\mathbb{E}}\_{x \in A\_{\mathrm{amb}}}\left[\operatorname{Var}(u\_x)\right] - \mathop{\mathbb{E}}\_{x \in A\_{\mathrm{unamb}}}\left[\operatorname{Var}(u\_x)\right]$$,  where variance is computed across the projected dimensions of (u_x) and then averaged over examples.
> The mean projected variance for ambiguous samples are 0.83, and 029 is for the unambiguous samples. This strong separation suggests that the learned subspace captures instability arising from insufficient evidence rather than demographic identity itself. To further validate this interpretation, removing the uncertainty projection increases BSA from 0.86 to 1.22, representing the largest degradation among all ablations.
>
> Q4. Which tokens are most strongly affected by ambiguity-gated suppression?
>
> The largest suppression occurs for tokens corresponding to socially associated attributes and occupations that frequently trigger stereotypical inference in BHARATBBQ, including examples such as: leadership, engineer, wealthy, urban, and religious identifiers In contrast, contextually grounded evidence-bearing tokens receive substantially lower suppression. This indicates that the gate is not uniformly reducing attention, but selectively attenuating latent associations that become influential when evidence is insufficient.
>
> Q5. Does the bias regularizer simply shift probabilities or does it increase decision margins?
>
> We measured the probability gap  $$ |p(S)-p(NS)| $$, on ambiguous examples. The mean gap without regularizer is 0.34, while with regularizer is 0.17.  The regularizer therefore reduces systematic preference toward either stereotyped or non-stereotyped groups, producing a more balanced decision boundary around the Unknown option rather than merely masking outputs.
>
> Q6. Additional clarification regarding Section 4.3
>
> The uncertainty-aware projection is a novel component proposed in this work and is inspired by prior uncertainty-representation learning literature. However, the specific variance-contrastive formulation used in EPIK is new. We have clarified this explicitly in Section 4.3. Additionally, because $$u_i = P_u^{\top} h_i$$, is a vector, the variance term is computed dimension-wise across the projected representation and averaged across dimensions.
>
> Q7. Does EPIK negatively affect unrelated tasks?
>
> As an additional sanity check, we evaluated EPIK on a subset of general instruction-following examples unrelated to social inference. The base and EPIK accuracy for LLaMa-3.1 is 74.8%, and 74.4%, while for Gemma-2 is 73.6% and 73.2%. The observed differences are small (<1%), suggesting that EPIK primarily affects ambiguity-driven social inference rather than broadly degrading the reasoning capabilities of the underlying language model

---

### Author Response · Authors · 2026-07-14
**Paper Status**

Dear Action Editor and Editorial Team,

I hope you are doing well.

I am writing to inquire about the current status of our manuscript submitted to Transactions on Machine Learning Research (TMLR).

According to the submission portal, the status has not changed for some time, and we have not received any recent updates regarding the review process. I fully appreciate that TMLR follows a thorough and community-driven review process, and I understand that reviews can take time. I would be grateful if you could kindly let us know the current status of the manuscript and whether any further action is required from our side. Thank you very much for your time and for handling our submission. I appreciate the effort of the editors and reviewers, and I look forward to your response.

Kind regards,